# Non-Invasive Radiofrequency Diathermy Neuromodulation Added to Supervised Therapeutic Exercise in Patellofemoral Pain Syndrome: A Single Blind Randomized Controlled Trial with Six Months of Follow-Up

**DOI:** 10.3390/biomedicines12040850

**Published:** 2024-04-11

**Authors:** Manuel Albornoz-Cabello, Alfonso Javier Ibáñez-Vera, Cristo Jesús Barrios-Quinta, Luis Espejo-Antúnez, Inmaculada Carmen Lara-Palomo, María de los Ángeles Cardero-Durán

**Affiliations:** 1Department of Physiotherapy, Universidad de Sevilla, 41004 Sevilla, Spain; 2Department of Health Sciences, Universidad de Jaén, 23009 Jaén, Spain; 3Andalusian Health Service, 41071 Sevilla, Spain; 4Department of Medical-Surgical Therapy, Universidad de Extremadura, 06006 Badajoz, Spain; luisea@unex.es (L.E.-A.);; 5Department of Nursing, Physical Therapy and Medicine, Universidad de Almería, 04120 Almería, Spain

**Keywords:** neuromodulation, radiofrequency therapy, knee pain, patellofemoral pain syndrome, diathermy, therapeutic exercise, physical therapy modalities

## Abstract

The evidence-based treatment of patellofemoral pain (PFP) suggests that therapeutic exercise (TE) focused on improving muscle strength and motor control be the main conservative treatment. Recent research determined that the success of the TE approach gets improved in the short term by the addition of neuromodulation via radiofrequency diathermy (RFD). As there is no follow up data, the objective of this research is to assess the long-term effects of adding RFD to TE for the pain, function and quality of life of PFP patients. To this aim, a single-blind randomized controlled trial was conducted on 86 participants diagnosed of PFP. Participants who met the selection criteria were randomized and allocated into either a TE group or an RFD + TE group. TE consisted of a 20 min daily supervised exercise protocol for knee and hip muscle strengthening, while RFD consisted of the application of neuromodulation using a radiofrequency on the knee across 10 sessions. Sociodemographic data, knee pain and lower limb function outcomes were collected. The RFD + TE group obtained greater improvements in knee pain (*p* < 0.001) than the TE group. Knee function showed statistically significant improvements in Kujala (*p* < 0.05) and LEFS (*p* < 0.001) in the RFD + TE group in the short and long term. In conclusion, the addition of RFD to TE increases the beneficial effects of TE alone on PFP, effects that remain six months after treatment.

## 1. Introduction

Patellofemoral pain (PFP) describes the presence of pain around or behind the patella during knee loading activities that trends toward chronicity [1,2]. Clinical practice guidelines recommend that therapeutic exercises be the main treatment (although there is no consensus about frequency and load), especially supervised exercises, as self-exercises have a reduced efficacy in contrast [3,4,5]. Furthermore, guidelines emphasize that PFP exercises must include both the knee and the hip joint and advise against the use of electrophysical agents such as laser, ultrasound and electrical stimulation [1,6]. However, it must be noted that electrophysical agents is a wide term that cannot be generalized as the mechanisms of the different techniques are not the same. Regarding to this, some poorly studied techniques, although very popular among physiotherapists, such as radiofrequency diathermy, have not been considered in clinical guidelines due to the lack of studies about them [1,6].

Neuromodulation via the use of radiofrequency diathermy consists of the non-invasive application of radiofrequency waves on biological tissues in order to increase their metabolism [7] and reduce pain, which improves their functioning [5]. This technique has shown efficacy in the treatment of knee symptoms in osteoarthritis [8] and PFP [5] among other musculoskeletal disorders [9,10,11,12,13]. The main difference between this technique and the use of other electrophysical agents is that radiofrequency waves allow for the deposit of energy in deeper tissues without heating the surface [14].

Supervised therapeutic exercises that includes the knee and hip joints are considered to be the most recommendable therapy for PFP [1,6,15]. However, there is an interest in adding other complementary techniques to this approach that could increase treatment efficacy by promoting tissue regeneration [16] and reducing the time of the intervention [1,6]. Due to the scarce evidence about the effects of neuromodulation using radiofrequency diathermy in PFP and the hypothesis that this technique could complement the effects of exercises by reducing pain and recovery time [5], the objective of this study is to investigate the effects of adding neuromodulation via radiofrequency diathermy to knee and hip exercises on the pain, function and quality of life of PFP patients.

## 2. Materials and Methods

### 2.1. Study Design

A single-blind randomized clinical trial was conducted. This trial was performed following the recommendations of the Helsinki Declaration, the good clinical practice rules and the regulations for research in humans, being drafted according to the CONSORT (Consolidated Standards of Reporting Trials) statement. The study was prospectively registered in ClinicalTrials.gov (ID number NCT05471089) and ethics committee approval was obtained (Ethics Committee of Hospital Virgen de la Macarena, Seville, Spain, approval code 2018Fisio01; date of approval 6 April 2019).

All participants were informed by one of the researchers about the study and their right to withdraw at any time without giving explanations before signing the informed consent.

### 2.2. Sample Size

Sample size calculation was based on the detection of (1) an improvement of 15% in self-perceived pain intensity [17], (2) a difference of >9 points in the LEFS score at the inter-group comparison after the treatment [18], and >10 points in the Kujala score. Considering a one-tail hypothesis, for ANOVA, repeated measures, a within-between interaction, an alpha value of 0.05, a desired power of 80%, and a medium effect size (r^2^ = 0.14), 42 participants were required per treatment group (G * Power, version 3.1.9.2). To increase the statistical power, a target of almost 80 participants was determined.

### 2.3. Participants

Three primary care physicians from a facility center in San José de la Rinconada, Seville (Spain) referred potential participants between October and December 2019. Participant selection was performed by a physician of the same facility center according to the following inclusion criteria: (1) subjects between 30 and 50 years old; (2) subjects referring to their anterior knee pain as being 3 or greater on a 10 cm Visual Analog Scale (VAS) for a minimum of eight weeks before the assessment or having anterior or retropatellar knee pain during at least three of the following activities: ascending/descending stairs, squatting, running, kneeling, jumping, and prolonged sitting; with any two of the following: a patellar compression, a palpation of the peripatellar region, and a resisted isometric knee extension when sitting [19], and (3) a score on the Personal Psychological Apprehension Scale (PPAS) that is below 45 [20]. Among the exclusion criteria, the following were established: (1) presenting contraindications to receiving a radiofrequency emission (tumor history, implanted pacemakers/electronic devices, episodes of thrombophlebitis or deep venous thrombosis, pregnancy, fever, infections or tuberculosis) [14,21], (2) having undergone knee treatment with corticoids or hyaluronic acid or platelet-rich plasma injections, and (3) participants with intra-articular knee pathology (>1 Kellgren-Lawrence Scale) [22], patellar instability, Osgood–Schlatter or Sinding-Larsen–Johansson syndrome. Patients currently involved in a medico-legal dispute were also excluded.

In the six months following selection for the study, in both groups, participants were allowed to use analgesic drugs with the aim of avoid introducing significant changes in their routine treatment, but were asked to record any drug intake (frequency and dosage) along the study in order to control for any significant change.

### 2.4. Randomization and Blinding

The researcher collecting data from participants was blinded to their group allocation. EPIDAT software in version 3.1 was used to perform the randomized allocation, considering a 1:1 split for the control (supervised exercises) and the experimental group (supervised exercise and neuromodulation using radiofrequency diathermy). Another researcher that was different from the one collecting data distributed the allocation to participants in opaque envelopes, ignoring their content. To ensure that participants were unaware of the treatment performed by the other group, the groups were given program instructions separately.

### 2.5. Outcomes

Sociodemographic data including age, weight, height, body mass index, fat mass, metabolic age and metabolic consumption were recorded with the help of a Body Composition Analyser TBF-400 (Tanita^®^, Tokyo, Japan) [23] and registered in a personal file prepared by the researchers.

Data related to pain were measured using the VAS. The VAS is a reliable, valid, responsive, and frequently used pain outcome measure. It consists of a bidirectional straight line with two labels located at either end of the line: “no pain” and “worst pain intensity”. The participant had to mark the pain experienced in the treated knee in the last 24 h. A 0 mm score would be interpreted as the participant having “no pain” and a 100 mm score as them having “extreme and insufferable pain” [24]. The minimal detectable change for the VAS is a reduction of 0.08 mm or a variation between 15–20% from the baseline score. The standard error of measurement (SEM) was 0.03 [25]. Measurements were performed at baseline, at the end of the last treatment session and six months after the last treatment.

Regarding knee function, two scales were used to assess functionality in participants. On one hand, there was their Kujala score [26], with a minimal clinically important difference of 9.5 points [27], and on the other hand, the Lower Extremity Functional Scale (LEFS), with a minimal clinically important difference of 9 points in the score [18]. The active flexion and extension range of motion was also measured with a conventional two-leg goniometer, obtaining an angular measurement. This instrument has shown good intra-tester (intraclass correlation coefficient (ICC) of 0.997 for flexion and 0.972–0.985 for extension) and inter-tester (ICC of 0.977–0.982 for flexion and 0.893–0.926 for extension) reliability [28]. Measurements were performed at baseline, at the end of the last treatment session and six months after the last treatment.

Finally, the PPAS was used at baseline to evaluate the apprehension of the subjects toward receiving electrical stimulation, which was one of the selection criteria, excluding subjects with scores below 45 points. This is a validated and reliable tool used for recognizing subjects whose apprehension toward electrical stimulation treatment could bias the results of the investigation [20].

### 2.6. Interventions

The interventions were carried out between October 2019 and March 2020, while the follow-up period continued until August 2020. Participants were allocated either into a control group or an intervention group. Both groups were supervised in completing knee and hip exercises for PFP by a physiotherapist, who was blind to group allocation. Due to the lack of consensus about these exercises [6], in this study, the protocol performed in the Andalusian Public Health Service was chosen, which consisted of concentric and eccentric strengthening exercises: three series of 20 s performing a bridge exercise for the hamstrings, three squat series of 20 repetitions for the quadriceps, three series of 20 s of the clam exercise for the gluteus medius and three series of 1 min of stretching exercises for the gastrocnemius and soleus. The whole set of exercises took 20 min approximately with a resting time of one minute between each of them, remembering not to exceed 3 mm on the VAS at any time during their performance. In order to supervise the exercises, all participants were assisted daily (from Monday to Friday) across three weeks to a local facility center, where a member of the research team took care of every set and exercise. After the treatment, patients were not asked to continue performing the exercises.

The experimental group received neuromodulation via radiofrequency diathermy in addition to the exercise protocol. Ten sessions of treatment were applied: daily during the first week (Monday to Friday), three times during the second week (Monday, Wednesday, Friday) and two times during the last week (Monday and Thursday). A monopolar dielectric device was used (ABD Modular^®^, Biotronic Advance Develops^®^, Granada, Spain) in a pulsed emission of 30 V that constantly varied from 640 kHz to 830 kHz to avoid an accommodation to phenomena (Figure 1). A dynamic application was performed with a continuous rotational and translational movement of the anterior surface of the knee being chosen. Due to monopolar dielectric transmission, five milliliters of almond oil were applied to improve the gliding of the applicator along the twelve-minute session [5]. The treatment was applied by a physiotherapist with over 10 years of experience in diathermy devices.

### 2.7. Statistical Analysis

A descriptive analysis was conducted for each variable. All results were reported as means and (standard deviation) considering 95% confidence intervals (95% CI). Normality tests were performed on the variables analyzed using the Shapiro–Wilk test. Next, the homogeneity of variance was assessed using Levene’s test. Bivariate dispersion graphics of the residual values observed from the expected valued were used to evaluate linearity. Between-group comparisons for both the demographic and the clinical data were performed using Student’s *t*-test for continuous variables, while Pearson’s chi-square test was chosen for categorical variables.

The groups were analyzed in a randomized fashion, and the intention-to-treat principle was followed in all analyses. We used a repeated-measures analysis of variance (ANOVA) 2 × 3 to investigate the differences in measurements. Group-per-time interactions* were assessed with a mixed analysis of variance (ANOVA), including the effect of time (at baseline, after 3 weeks of treatment and after the 6 month follow-up) as an intra-subject factor while group effects (MDR group and exercise group) were considered as inter-subject factors. Post hoc comparisons (Bonferroni) were performed for significant effects. Eta squared (η^2^) was used to calculate effect size (small when 0.01 ≤ η^2^ ≤ 0.06; medium when 0.06 ≤ η^2^ > 0.14; large when η^2^ > 0.14). The data was analyzed using PASW Advanced Statistics (SPSS Inc., Chicago, IL, USA), version 24.0. Statistical significance was determined at *p* < 0.05.

## 3. Results

A total of 90 subjects, with ages between 33 and 49 were selected for the trial. After the inscription phase, the final sample included 86 participants (n = 86) (Figure 2), 34 men and 52 women with a mean age of 42 (SD 4.27) years old. Four participants refused to participate in the study and another two did not meet the inclusion criteria as they referred to pain only occurring in the last two weeks. Out of the knees treated, 46 were right knees (54%) while the remaining 40 were left (46%).

The baseline demographic characteristics (age, sex, high, weight, body mass index), the clinical ones (level of functionality and self-perceived pain) and the ROM (flexion and extension) are shown in Table 1. No participants reported changes in drug intake routines during the time of the study. Statistically significant differences were not found at an inter-group level for any of the variables (*p* > 0.05 for all).

Table 2 includes the baseline, final and six-month follow-up measurements, as well as the between-groups and intra-group mean differences. Statistically significant differences were found to favor the experimental group in pain perception (VAS F_1, 84_ = 109.74 [*p* = 0.000]) η^2^ = 0.56) in knee disability (LEFS: F_1, 84_ = 16.78 [*p* = 0.000] η^2^ = 0.17 and KUJALA: F_1, 84_ = 13.11 [*p* = 0.001] η^2^ = 0.14) and in their range of movement (Flexion: F_1, 84_ = 49.85 [*p* = 0.000] η^2^ = 0.372 and Extension: F_1, 84_ = 0.95 [*p* = 0.331] η^2^ = 0.01). Figure 3 shows the between-group differences in the intensity of pain perceived by participants. No statistically significant differences were found between the before- and after-treatment measurements regarding the use of basic analgesic drugs. Finally, it must be noted that no side effects were observed.

## 4. Discussion

PFP still poses a challenge for healthcare providers. Although an important effort has been made investigating its etiology and treatment, no consensus has been reached among experts regarding frequency and load in exercises [6]. Moreover, clinical guidelines advise against the use of electrophysical agents [1,6], a recommendation that must be revised in light of the results of this study and the fact that under the term “electrophysical agents” underlie different techniques with different mechanisms whose effects could differ. In this sense, a recent systematic review by Fari et al. pointed out the relevance of analyzing, in a meticulous manner, the efficacy of radiofrequency use for musculoskeletal pain according to the application modality and body region [29].

This study is, to our knowledge, the first to study the effects of adding radiofrequency diathermy to supervised exercises for the treatment of PFP with a long-term follow-up. Other studies have determined that the addition of radiofrequency diathermy to home exercises increases their benefits for the pain and functioning with PFP [5]. However, studies support that home exercises are not as effective as supervised ones [3,30], probably due to the fact that treatment adherence gets reduced without supervision [4]. For this reason, a study comparing the effects of radiofrequency diathermy with the most effective exercises was needed. According to this study, non-invasive neuromodulation via radiofrequency diathermy is an effective complement to exercise treatment for pain in PFP, obtaining statistically significant improvements with large effect sizes for all variables at short and long terms over exercise alone (Table 2). Regarding the VAS, our results are consistent with the ones reported by García-Marín et al. [14] after applying the same number of sessions (a total of 10 sessions, one session per day, from Monday to Friday for two weeks) of radiofrequency therapy in the postoperative phase of knee arthroplasty. The improvements observed in our study were superior both in the group receiving radiofrequency (48.4 mm vs. 34.4 mm) as well as in the control group (14.2 mm vs. 9.2 mm), being above the established MDC [25]. These between-group differences could be explained by (1) the sample size of each of the groups, (2) the periodicity of treatment administration and (3) the underlying neurophysiological mechanism after surgery. Concerning the observed changes at the six-month follow-up in the RFD group, the pain intensity on the VAS showed a higher percentual reduction compared with that reported by Giles et al. after applying quadriceps-strengthening exercise with and without blood flow restriction (49.5%/50.2% vs. 88.2%) in subjects diagnosed with PFP [31]. Considering the moderate pain sensitization observed in subjects with PFP [32], future studies must assess if pain sensitization may be amenable to treatment through non-invasive radiofrequency diathermy neuromodulation combined with exercise therapy.

Regarding the exercise control group, despite the observed improvements (Table 2), it must be noted that from a clinical point of view, these could be considered poor results. This could point out that exercise treatments require larger time interventions [33], which could get reduced by the use of neuromodulation via radiofrequency diathermy. In this line, the between-group differences favoring the experimental diathermy group despite the short time intervention could be interpreted as this technique highly reducing the time for recovery in PFP, which is a relevant finding. It must be considered that other complementary therapies such as taping techniques being added to knee exercises also seem to obtain poorer results than the addition of radiofrequency diathermy [34]. These results agree with the ones obtained in knee osteoarthritis by Kumaran and Watson, where pain improvements achieved with radiofrequency diathermy treatment reached four points on the VAS [8]. Their long-term results are lower compared with the ones of the present study, which could be due to the point that knees in osteoarthritis present higher tissue degeneration than in PFP [16]. Both the present study with supervised exercise and the one of Albornoz-Cabello et al. (2020) with home exercises agree about the effectiveness of adding RFD to knee exercises, obtaining similar results [5]. Based on this, we could not recommend supervised exercises over training patients in home exercises. Regarding knee function, clinical guidelines support that therapeutic knee exercises obtain successful results [1,6]. A time-per-group interaction was observed for the knee function variables (LEFS % and Kujala %), where significant differences were obtained between groups after the intervention and six months of follow-up with better results occurring for the RFD group (Table 2). These results are similar to those reported by Garcia-Marin et al. after applying the WOMAC questionnaire to subjects during the postoperative phase of knee arthroplasty [13]. The modest improvements observed in this study in the control group could point out that more than a month of intervention with therapeutic knee exercises is needed to obtain the appropriate results. The pain relief experienced by participants of radiofrequency diathermy group could have facilitated their exercise performance, allowing them to obtain greater improvements in knee function, their range of movement and lower limb function, as was observed in previous studies [35]. This explanation agrees with the study of Kumaran and Watson (2019), in which 19 points of improvement on the WOMAC were obtained in knee osteoarthritis patients treated with radiofrequency diathermy added to exercises while poor results were observed in the control group [8]. Moreover, the changes observed at six months (Table 2) showed improvements of 41.8% in the RFD group and 20.7% in the control group. These percentual changes were superior to those reported by Giles et al. These authors obtained an improvement of 14.7% and 18.3% in knee function measured with the Kujala questionnaire after applying quadricep-strengthening exercise combined with and without blood flow restriction, respectively, in subjects diagnosed with PFP [31]. These differences observed between studies could be due to the poorer baseline functional status of the participants of the present study. Finally, it is a fact that superior significant improvements were observed for this variable in the group receiving RFD and exercise therapy after six months of follow-up, so neuromodulation via RFD treatment should be considered a recommendable complement to reduce the time of recovery in a therapeutic exercise approach for PFP. Although a return of symptoms was expected in the follow-up, the benefits in pain and knee function observed as well as the absence of adverse reactions seems to last in time, almost continuing after six months. In this sense, the authors believe that it is clinically relevant that the participants were controlled with some of the inclusion criteria of the study. Specifically, the fact that we measured personal psychological apprehension as a psychosocial covariate in both groups (Table 1) and that the participants had prior knowledge of the sensory responses during the application of the RFD procedure are controlling factors for possible adverse reactions.

### 4.1. Study Limitations

This study presents several limitations: first of all, the lack of consensus among exercises for PFPS makes it very difficult to compare whether the efficacy of the treatment is higher than others or not. Another important point is that the short duration of the treatment probably did not allow for observing greater improvements in the exercise group; however, this was not the objective of the study and this has revealed the potential of neuromodulation via radiofrequency diathermy to accelerate function recovery. In addition, the study lacked a sham group to identify a potential placebo effect, and a wait-list group to reflect the natural course of the condition. It also must be considered that the sample was recruited in the same place, which could involve a representation bias. Future studies should concern delimitating the effects of RFD and exercise in separate groups. Finally, the blinding of participants was not possible, and the duration of treatment sessions slightly differed between groups, which could be a source of bias.

### 4.2. Clinical Implications

The benefits observed in both groups could be partially due to the fact that the multimodal program was supervised by qualified personnel, achieving greater benefits and adherence than when exercise is performed without professional supervision. This innovative procedure allows for improvements in pain severity, function and range of motion to be maintained for at least six months. Considering the effectiveness of the patient education in subjects with knee osteoarthritis [36], future studies are needed to analyze the clinical impact of these combined conservative treatments (education, RFD and exercise therapy) as assessed by the number needed to treat.

## 5. Conclusions

The addition of neuromodulation via radiofrequency diathermy to a knee and hip exercise program of three weeks obtained greater improvements in patellofemoral pain than the knee and hip exercises alone.

## Figures and Tables

**Figure 1 biomedicines-12-00850-f001:**
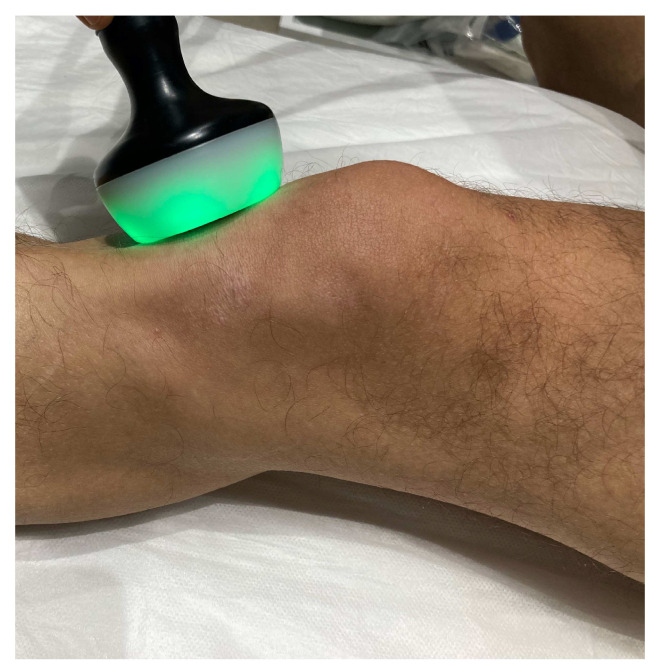
Radiofrequency diathermy’s application on a knee.

**Figure 2 biomedicines-12-00850-f002:**
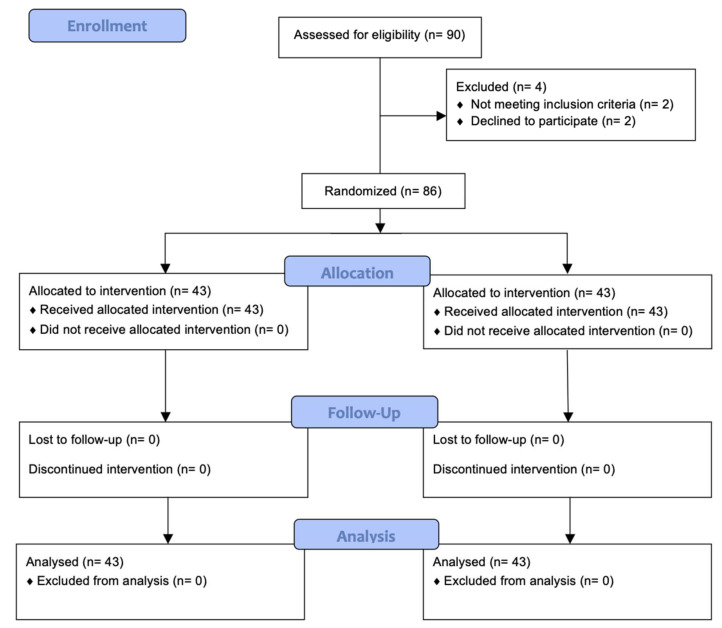
Flow diagram of participants.

**Figure 3 biomedicines-12-00850-f003:**
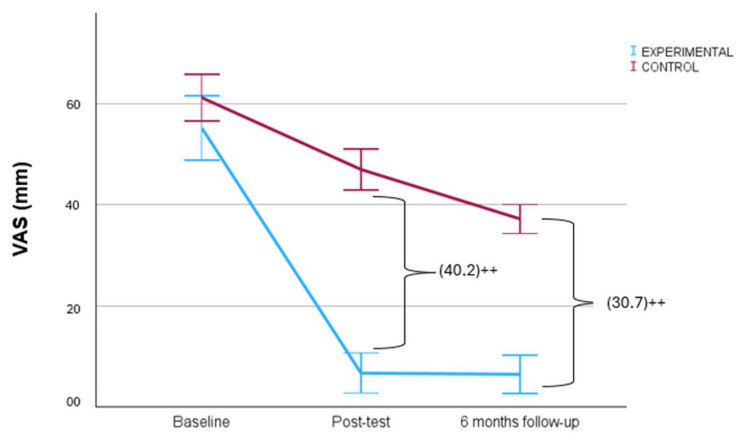
Between-group differences in pain intensity perceived along the treatments. ++: *p* < 0.001.

**Table 1 biomedicines-12-00850-t001:** Baseline characteristics of participants in the study groups.

	Total Sample (n = 86)	RF Group (n = 43)	Control Group (n = 43)	*p* Value *
Mean age (years)	42 (4.27)	42 (4.39)	43 (4.1)	0.159
Height (cm)	167 (10.48)	166 (11.21)	167 (9.80)	0.683
Weight (kg)	78.3 (14.85)	77.7 (14.01)	78.9 (15.78)	0.712
Body Mass Index	28.2 (4.93)	28.3 (5.21)	28.2 (4.69)	0.929
Fat mass (%)	31.5 (10.30)	31.6 (10.63)	31.5 (10.08)	0.968
Metabolic age (years)	51 (19.46)	49 (20.71)	53 (18.15)	0.361
BMR (KJ)	6716 (1407.18)	6640 (1282.65)	6792 (1533.13)	0.620
LEFS (%)	54 (18.27)	57 (20.43)	51 (15.38)	0.104
KUJALA (%)	53 (18.18)	55 (21.07)	51 (14.81)	0.390
VAS (mm)	58 (18.2)	55 (20.6)	61 (15.0)	0.127
Flexion (°)	113 (12.37)	115 (11.67)	112 (13.06)	0.363
Extension (°)	0.7 (1.95)	0.9 (2.25)	0.6 (1.62)	0.412
PPAS	27 (8.01)	26 (7.07)	30 (8.11)	0.052

Data are reported as means (standard deviation). PPAS, Personal Psychological Apprehension Scale; RF, radiofrequency diathermy; *** Between-groups statistical significance (one factor ANOVA).

**Table 2 biomedicines-12-00850-t002:** Baseline, immediate post-test and follow-up mean score changes of knee pain and lower extremity function.

	Group	Baseline	Immediate Post-Test	Follow-Up	Baseline/Immediate Post-Test Differences	Between-Group Mean Changes on the Immediate Post-Test	Immediate Post-Test/Follow-Up Differences	Between-Group Mean Changes at the Follow-Up
LEFS (%)	RFD	57 ± 20.43	74 ± 14.71	77 ± 15.38	17 (13–20) **	18 (11–23) ^††^	3 (1–4) *	16 (9–21) ^††^
	Control	51 ± 15.39	56 ± 12.83	61 ± 12.03	5 (4–7) **		5 (4–6) **	
KUJALA (%)	RFD	55 ± 21.07	74 ± 13.61	78 ± 14.09	19 (14–23) **	17 (11–23) ^††^	4 (2–6) *	14 (9–20) ^††^
	Control	53 ± 14.9	57 ± 15.1	64 ± 12.6	4 (3–5) **		7 (4–8) **	
VAS (mm)	RFD	55.2 ± 20.6	6.7 ± 12.8	6.5 ± 12.3	48.4 (41.9–54.9) **	40.2 (45.8–34.6) ^††^	0.2 (−1.9–2.4)	30.7 (35.4–26.0) ^††^
	Control	61.1 ± 15.0	46.9 ± 13.2	37.2 ± 9.3	14.2 (11.5–16.8) **		9.7 (5.9–13.5) **	
Flexion (°)	RFD	115 ± 11.6	133 ± 7.6	134 ± 5.6	18 (16–20) **	20 (15–24) ^††^	1 (0.5–2.1)	24 (20–28) ^††^
	Control	112 ± 13.0	115 ± 10.2	115 ± 7.5	3 (0.4–4.2) *		0.7 (−2.7–1.4)	
Extension (°)	RFD	0.9 ± 2.2	0.1 ± 0.7	0 ± 0	0.8 (0.1–1.5) *	0.6 (0.3–1.2)	0.1 (−0.1–0.3)	0.7 (0.3–1.3) ^†^
	Control	0.6 ± 1.62	0.7 ± 2.08	0.7 ± 2.07	0.1 (−0.3–0.2)		0 (−0.1–0.1)	

Data are reported as mean ± SD or [95% confidence level]. RFD: neuromodulation via radiofrequency diathermy; * indicates statistically significant intragroup differences (*p* < 0.05); ** indicates statistically significant intragroup differences (*p* < 0.001); ^†^ indicates statistically significant between-group differences (*p* < 0.05); ^††^ indicates statistically significant between-group differences (*p* < 0.001).

## Data Availability

Data of this study are available upon reasonable request to the corresponding author.

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
