# Peer review of "Non-Invasive Radiofrequency Diathermy Neuromodulation Added to Supervised Therapeutic Exercise in Patellofemoral Pain Syndrome: A Single Blind Randomized Controlled Trial with Six Months of Follow-Up"

_biomedicines, 2024, doi:10.3390/biomedicines12040850_

Round 1

Reviewer 1 Report

Comments and Suggestions for Authors

Dear authors,

your paper is very interesting, but many concerns have to be addressed.

Abstract is clear and concise.

Keywords: you should use some keywords different from those used in the title in order to improve the visibility of your paper.

Introduction is well written.

Methods: this section should be carefully corrected. In the title you said that this is a single blind trial, but in the methods you declared that it is a double blind trial. Please, clarify this aspect and specify clearly how the blinded study design was granted. 

You enrolled subjects between 30 and 50 years old without radiological findings of osteoarthritis. Which radiological findings did you admit/require to recruit a patient with patellofemoral syndrome?

Then, from a methodological point of view it is trivial to understand that a group which received two treatments had better outcomes if compared to the group which underwent therapeutic exercise alone. Why did not you carried out a four groups study, including also a group destinated to radiofrequency alone and another one destinated to placebo radiofrequency using sham? Explain your choice. In this sense the current literature on RF could help you, so I suggest the following reference:

Farì, G., de Sire, A., Fallea, C., Albano, M., Grossi, G., Bettoni, E., Di Paolo, S., Agostini, F., Bernetti, A., Puntillo, F., & Mariconda, C. (2022). Efficacy of Radiofrequency as Therapy and Diagnostic Support in the Management of Musculoskeletal Pain: A Systematic Review and Meta-Analysis. Diagnostics (Basel, Switzerland)12(3), 600. https://doi.org/10.3390/diagnostics12030600

Then, from an ethical point of view and in accordance with the International pain management guidelines, you should guarantee to all the patients who undergo a rehabilitation treatment also the possibility to use some drugs in case of persistent pain during the therapy. Did you allow patients to use analgesics such as paracetamol and to fill an intake diary in this sense? If no, why? Please explain.

Results: please reconsider the graphical aspect of the table 2. It is not clear and it is not possible to understand, at the moment, how intervention group was better than the control one in the outcomes.

Discussion should be briefly integrated giving your findings a more clinical lapel. In fact, you had the merit to propose an integrated rehabilitation strategy using radiofrequency, you should highlight this aspect and the importance of combined treatment in rehabilitation. To do that, I suggest the following reference:

Mariconda, C., Megna, M., Farì, G., Bianchi, F. P., Puntillo, F., Correggia, C., & Fiore, P. (2020). Therapeutic exercise and radiofrequency in the rehabilitation project for hip osteoarthritis pain. European journal of physical and rehabilitation medicine56(4), 451–458. https://doi.org/10.23736/S1973-9087.20.06152-3

I hope these suggestions could help ypu to improve the quality of your interesting paper.

Best regards and good luck

Author Response

Dear reviewer, 

Thank you very much for your time and effort. Your
suggestions and comments have improved the quality of our work. Please find the answers to your comments below each of them:

-Dear authors, your paper is very interesting, but many concerns have to be addressed.

A: Thank you.

-Abstract is clear and concise.

A: Thank you.

Keywords: you should use some keywords different from those used in the title in order to improve the visibility of your paper.

A: Thank you. We have checked and modified the keywords.

-Introduction is well written.

A: Thank you.

-Methods: this section should be carefully corrected. In the title you said that this is a single blind trial, but in the methods you declared that it is a double blind trial. Please, clarify this aspect and specify clearly how the blinded study design was granted. 

A: We totally agree with the reviewer in this point. This has been corrected in Methods section (line 65).

-You enrolled subjects between 30 and 50 years old without radiological findings of osteoarthritis. Which radiological findings did you admit/require to recruit a patient with patellofemoral syndrome?

A: Thank you for your suggestion, which has improved the study in this aspect. We have clarified this issue in exclusion/inclusion criteria (lines 88-102) according to Kellgren-Lawrence Scale:

Kellgren JH, Lawrence JS (1957) Radiological assessment of osteo-arthrosis. Ann Rheum Dis 16:494–502), patellar instability, Osgood-Schlatter or Sinding-Larsen-Johansson syndrome.

Participant

-Then, from a methodological point of view it is trivial to understand that a group which received two treatments had better outcomes if compared to the group which underwent therapeutic exercise alone. Why did not you carried out a four groups study, including also a group destinated to radiofrequency alone and another one destinated to placebo radiofrequency using sham? Explain your choice. In this sense the current literature on RF could help you, so I suggest the following reference:

A: Thank you very much for your comment. Our design selection was made based on a review of the scientific literature. Recent studies were design in a similar way fixing therapeutic exercise as gold standard treatment for patellofemoral pain without osteoarthritis:

Collins, N.J.; Barton, C.J.; Van Middelkoop, M.; Callaghan, M.J.; Rathleff, M.S.; Vicenzino, B.T.; Davis, I.S.; Powers, C.M.; Macri, E.M.; Hart, H.F.; et al. 2018 Consensus Statement on Exercise Therapy and Physical Interventions (Orthoses, Taping and Manual Therapy) to Treat Patellofemoral Pain: Recommendations from the 5th International Patellofemoral Pain Research Retreat, Gold Coast, Australia, 2017. Br J Sports Med 2018, 52, 1170–1178, doi:10.1136/bjsports-2018-099397

Van Der Heijden, R.A.; Lankhorst, N.E.; Van Linschoten, R.; Bierma-Zeinstra, S.M.A.; Van Middelkoop, M. Exercise for Treating Patellofemoral Pain Syndrome: An Abridged Version of Cochrane Systematic Review. Eur J Phys Rehabil Med 2016, 52, 110–133.

This fact has influenced the posterior studies, adding to therapeutic exercise different treatments such as radiofrequency in knee joint replacement (García-Marín, M.; Rodríguez-Almagro, D.; Castellote-Caballero, Y.; Achalandabaso-Ochoa, A.; Lomas-Vega, R.; Ibáñez-Vera, A.J. Efficacy of Non-Invasive Radiofrequency-Based Diathermy in the Postoperative Phase of Knee Arthroplasty: A Double-Blind Randomized Clinical Trial. J. Clin. Med. 2021, 10, 1611. https://doi.org/ 10.3390/jcm10081611) or in knee osteoarthritis (Kumaran, B.; Watson, T. Treatment Using 448 KHz Capacitive Resistive Monopolar Radiofrequency Improves Pain and Function in Patients with Osteoarthritis of the Knee Joint: A Randomised Controlled Trial. Physiotherapy (United Kingdom) 2019, 105, 98–107, doi:10.1016/j.physio.2018.07.004).

Other authors added to therapeutic exercise another therapeutical approaches such as blood Flow restriction therapy in patelloemoral pain syndrome (Giles L, Webster KE, McClelland J, Cook JL. Quadriceps strengthening with and without blood flow restriction in the treatment of patellofemoral pain: a double-blind randomised trial Br J Sports Med 2017;51:1688–1694. doi:10.1136/bjsports-2016-096329).

However, up to our knowledge there is not a study about the inmediate and long term effects of radiofrequency in patelofemoral pain. This fact added to the role of exercise as gold standard in the approach of knee pain (Collins, N.J.; Barton, C.J.; Van Middelkoop, M.; Callaghan, M.J.; Rathleff, M.S.; Vicenzino, B.T.; Davis, I.S.; Powers, C.M.; Macri, E.M.; Hart, H.F.; et al. 2018 Consensus Statement on Exercise Therapy and Physical Interventions (Orthoses, Taping and Manual Therapy) to Treat Patellofemoral Pain: Recommendations from the 5th International Patellofemoral Pain Research Retreat, Gold Coast, Australia, 2017. Br J Sports Med 2018, 52, 1170–1178, doi:10.1136/bjsports-2018-099397) and its combination with other modalities for musculoskeletal pain  (Farì, G., de Sire, A., Fallea, C., Albano, M., Grossi, G., Bettoni, E., Di Paolo, S., Agostini, F., Bernetti, A., Puntillo, F., & Mariconda, C. (2022). Efficacy of Radiofrequency as Therapy and Diagnostic Support in the Management of Musculoskeletal Pain: A Systematic Review and Meta-Analysis. Diagnostics (Basel, Switzerland)12(3), 600. https://doi.org/10.3390/diagnostics12030600), determined our decision regarding the desing of the trial. Nonetheless, we have added in limitations a recommendation about performing future studies with four arms in order to delimitate the effect of each intervention (lines 324-325)

Then, from an ethical point of view and in accordance with the International pain management guidelines, you should guarantee to all the patients who undergo a rehabilitation treatment also the possibility to use some drugs in case of persistent pain during the therapy. Did you allow patients to use analgesics such as paracetamol and to fill an intake diary in this sense? If no, why? Please explain.

A: Thanks for your question, which has helped us to clarify such a crucial aspect. Yes, basic drug intake was allowed (lines 103-106) as well as participant’s daily activities in order to avoid changes in their common activities, obtaining a realistic scenario. Any change related to drug intake has been described (lines 199-200).

Results: please reconsider the graphical aspect of the table 2. It is not clear and it is not possible to understand, at the moment, how intervention group was better than the control one in the outcomes.

A: We totally agree the reviewer in this point. A new figure (figure 3) has been designed about changes observed in the primary outcome intensity of perceived pain (lines 210-211).

Discussion should be briefly integrated giving your findings a more clinical lapel. In fact, you had the merit to propose an integrated rehabilitation strategy using radiofrequency, you should highlight this aspect and the importance of combined treatment in rehabilitation. To do that, I suggest the following reference:

Mariconda, C., Megna, M., Farì, G., Bianchi, F. P., Puntillo, F., Correggia, C., & Fiore, P. (2020). Therapeutic exercise and radiofrequency in the rehabilitation project for hip osteoarthritis pain. European journal of physical and rehabilitation medicine56(4), 451–458. https://doi.org/10.23736/S1973-9087.20.06152-3

A: Thank you for the comment. We have improved the Discussion section according to the suggestion of including comparisons with previous studies (lines 230-233, 243-258, 279-283, 289, 292-301)

I hope these suggestions could help you to improve the quality of your interesting paper.

A: Thanks a lot for your time and effort.

Best regards and good luck

Reviewer 2 Report

Comments and Suggestions for Authors

The authors have developed a well-conducted and well-written study with the aim of investigating the effects of adding radiofrequency diathermy with neuromodulation to knee and hip exercises on pain, function, and quality of life in patients with patellofemoral pain.

However, I would like to make a few observations before recommending their work for publication.

1. I request the authors to organize section 2.1 appropriately, and to detail the blinding process in another section.

2. There is an interesting study on percutaneous technique that I recommend the authors to discuss, in the light of the results obtained with radiofrequency: DOI: 10.52586/5017

3. Did the referring physicians diagnose osteoarthritis? Please specify the medical diagnosis of PFS

4. There is an interesting study on educational intervention that I recommend the authors to discuss, in light of the results obtained with exercise and radiofrequency: DOI: 10.3390/ijerph19106194

5. Could you add a section on "Clinical Implications"?

Comments on the Quality of English Language

No comments

Author Response

Dear Editor,

Thank you very much for your time and effort. Your comments have increase the quality of our work. Please find below each question its answer:

The authors have developed a well-conducted and well-written study with the aim of investigating the effects of adding radiofrequency diathermy with neuromodulation to knee and hip exercises on pain, function, and quality of life in patients with patellofemoral pain.

A: We want to thank the reviewer for the suggestion, that have improved the quality of our manuscript.

However, I would like to make a few observations before recommending their work for publication.

  1. I request the authors to organize section 2.1 appropriately, and to detail the blinding process in another section.

A: Thank you for the suggestion, it has been solved (lines 65-71 and 108-114)

  1. There is an interesting study on percutaneous technique that I recommend the authors to discuss, in the light of the results obtained with radiofrequency: DOI: 10.52586/5017

A: We really thank the reviewer for this recommendation. However, after reading this case series study we are not sure about its suitability for the discussion of the present manuscript, as it is about a minimally invasive approach (instead of a non-invasive like ours) based on a different electrical signal and applied to patellar tendinopathy and not to osteoarthritis or patellofemolar pain syndrome. We really appreciate the suggestion, but we think it does not fit with our topic. Anyway, we will consider it for future related research.

  1. Did the referring physicians diagnose osteoarthritis? Please specify the medical diagnosis of PFS

A: Thank you very much for this point, it has been clarified (line 87)

  1. There is an interesting study on educational intervention that I recommend the authors to discuss, in light of the results obtained with exercise and radiofrequency: DOI: 10.3390/ijerph19106194

A: Thank you very much, it has been included (line 330)

  1. Could you add a section on "Clinical Implications"?

A: Thank you very much for this suggestion. The new section has been included (lines 325-332)

Round 2

Reviewer 1 Report

Comments and Suggestions for Authors

Thank you for the efforts to improve the quality of Your paper.

No further corrections are needed.

best regards

Reviewer 2 Report

Comments and Suggestions for Authors

The authors have improved the previous version of their manuscript with the current version, so I recommend its publication.

Congratulations

Comments on the Quality of English Language

No comments